# Taking a Closer Look: Clinical and Histopathological Characteristics of Culture-Positive versus Culture-Negative Pulmonary Mucormycosis

**DOI:** 10.3390/jof8040380

**Published:** 2022-04-08

**Authors:** Amy Spallone, Cesar A. Moran, Sebastian Wurster, Dierdre B. Axell-House, Dimitrios P. Kontoyiannis

**Affiliations:** 1Department of Infectious Diseases, Infection Control and Employee Health, University of Texas MD Anderson Cancer Center, Houston, TX 77030, USA; aspallone@mdanderson.org (A.S.); stwurster@mdanderson.org (S.W.); 2Department of Pathology, University of Texas MD Anderson Cancer Center, Houston, TX 77030, USA; cesarmoran@mdanderson.org; 3Division of Infectious Diseases, Houston Methodist Research Institute, Houston Methodist Hospital, Houston, TX 77030, USA; dbaxell-house@houstonmethodist.org

**Keywords:** mucorales, pulmonary mucormycosis, hematologic malignancy, transplantation, fungal culture, histopathology

## Abstract

The cultural recovery of Mucorales from hyphae-laden tissue is poor, and the clinical implications of culture positivity are scarcely studied. Therefore, we compared clinical and histopathological characteristics of culture-positive and culture-negative histology-proven pulmonary mucormycosis cases among cancer patients. Histology specimens were blindly reviewed by a thoracic pathologist and graded on four histopathologic features: hyphal quantity, tissue necrosis, tissue invasion, and vascular invasion. Twenty cases with a corresponding fungal culture were identified; five were culture-positive, and fifteen were culture-negative. Although no statistically significant differences were found, culture-positive patients were more likely to exhibit a high burden of necrosis and have a high burden of hyphae but tended to have less vascular invasion than culture-negative patients. In terms of clinical characteristics, culture-positive patients were more likely to have acute myeloid leukemia (60% vs. 27%, *p* = 0.19), a history of hematopoietic cell transplant (80% vs. 53%, *p* = 0.31), severe lymphopenia (absolute lymphocyte count ≤ 500/µL, 100% vs. 73%, *p* = 0.36), and monocytopenia (absolute monocyte count ≤100/µL, 60% vs. 20%, *p* = 0.11). Forty-two-day all-cause mortality was comparable between culture-positive and culture-negative patients (60% and 53%, *p* = 0.80). This pilot study represents the first comprehensive histopathological scoring method to examine the relationship between histopathologic features, culture positivity, and clinical features of pulmonary mucormycosis.

## 1. Introduction

Mucormycosis (MCR) is the second most common cause of invasive mold infections in patients with hematologic malignancies (HMs) and recipients of hematopoietic cell transplants (HCTs) [1,2]. Invasive MCR carries a high mortality rate due to several factors, such as the absence of immune reconstitution, limited therapeutic options, and delayed diagnosis [2]. Often, routine laboratory methods fail to effectively recover Mucorales from tissue specimens despite evidence of large amounts of hyphae observed in histopathology. This poor recovery is thought to be from a failure of current culture methods’ ability to mimic physiologic conditions found in hyphae-laden infected tissue. Specifically, culture methods fail to provide optimal temperature and anaerobic conditions to facilitate the growth of stressed molds [2,3,4]. This was demonstrated in a case series from the University of Texas MD Anderson Cancer Center (MDACC), which examined 24 cases of MCR where the sensitivity of cultures in detecting Mucorales from specimens was low and often a preterminal sign [2]. A similar discrepancy has been observed in septate molds, where up to 60% of all histology-proven, septate hyphae-laden tissue from surgical biopsy and autopsy specimens failed to grow in culture [5].

To our knowledge, no prior study has correlated the microbiological recovery of MCR in culture with the histopathological and clinical features of cases of pulmonary MCR. To that end, we aimed to determine if specific histopathologic characteristics of pulmonary MCR were more likely to predict the growth of MCR in culture, were associated with higher rates of mortality, or were associated with specific clinical characteristics among patients with HMs and recipients of HCTs.

## 2. Materials and Methods

### 2.1. Inclusion and Exclusion Criteria

We retrospectively reviewed cases of histopathologically proven pulmonary MCR between April 2000 and April 2021 at MDACC among patients with HMs and HCTs. For inclusion, the pathology report on the pulmonary specimen had to specify that the hyphae present were consistent with Mucorales morphology without suggesting an alternative or additional non-MCR mold diagnosis. Furthermore, the type of specimen was restricted to surgical biopsies and autopsy specimens to ensure consistency during the evaluation of histopathologic features. Lastly, the histopathologic specimen had to have a corresponding tissue fungal culture collected in the same procedural time and space. Detailed inclusion and exclusion criteria are summarized in Table 1.

### 2.2. Chart Review

Clinical characteristics collected included age at diagnosis, which was defined as the date of histopathology collection, underlying malignancy, transplant history and type, history of graft-versus-host-disease (GVHD), recent antifungal exposure within 30 days preceding diagnosis, corticosteroid use totaling ≥600 mg prednisone dose equivalent in the 30 days preceding diagnosis, use of immunosuppressive agents other than steroids, and history of diabetes mellitus. Lab abnormalities at the time of diagnosis were also accounted for and included hyperglycemia (defined as blood glucose >200 mg/dL), malnutrition (defined as serum albumin ≤3 g/dL), renal failure (defined as serum creatinine >2.5 mg/dL), neutropenia (defined as absolute neutrophil count ≤500 cells/µL), lymphopenia (defined as absolute lymphocyte count ≤500 cell/µL), and monocytopenia (defined as absolute monocyte count ≤100 cells/µL).

### 2.3. Fungal Culture Processing and Incubation

Lung tissue specimens submitted for fungal culture were processed according to our microbiology laboratory’s protocols. Lung tissue was ground for 4–5 strokes in 5 mL of Tryptic Soy Broth (TSB) using a disposable tissue grinder with sand, and one mL was directed to fungal culture. Media utilized included two Emmons sabouraud dextrose agar (SDA) plates, one SDA slant, and one brain heart infusion agar slant with chloramphenicol and gentamicin supplements (BBL, Becton Dickinson, Sparks, MD, USA). Fungal media were placed at 35 °C, except the SDA slant, which was incubated at 25 °C and placed into a 5% CO_2_ environment immediately following setup. All fungal cultures were incubated for four weeks. In earlier studies, higher recovery of Aspergillus and Mucorales was achieved using the rapid transfer to 35 °C incubation [3,4]. Type of media, media supplements, and grinding versus mincing were also studied and showed no significant effects on the recovery of the organisms in culture [3,4].

### 2.4. Assessment of Histopathological Features

Identified cases underwent review by a thoracic pathologist (C.A.M.) blinded to the corresponding fungal culture results. All pulmonary specimen slides (i.e., hematoxylin and eosin (H&E) stain and Grocott-Gomori’s methenamine silver (GMS) stain) were reviewed in each case. Each specimen was quantitatively assessed at a set objective for four key histopathologic features: the amount of hyphae (20× objective), degree of tissue invasion (10× objective), degree of necrosis (10× objective), and degree of vascular invasion (20× objective). A representative example is shown in Figure 1a,b. These features were then graded on the amount of each feature present on the slide as a percentage of the slide, which ranged from “none” to “extensive” (>75–100%). An example scorecard is provided in Figure 1c.

### 2.5. Statistical Analyses

Fisher’s exact and Chi-square tests were used for univariate comparisons of clinical features in culture-positive and culture-negative patients. In addition, distributions of histopathological features in culture-positive versus culture-negative patients were compared using the Mann–Whitney U test. A *p*-value below 0.05 was considered significant. Data tabulation, analysis, and visualization were performed using GraphPad Prism v10 and Microsoft Excel.

## 3. Results

We identified a total of 20 patients (Table 2) with histology-proven pulmonary MCR whose diagnostic specimens met the inclusion criteria outlined above (Section 2.1). Most of these patients had underlying leukemia (n = 16, 80%), with acute myeloid leukemia being the most common HM in our cohort (n = 7, 35%, Table 2). Twelve patients (60%) had a history of HCTs (eleven allogeneic, one autologous). Seven out of the eleven allogenic HCT recipients had a history of graft-versus-host disease (64%). All but one patient had received recent antifungal prophylaxis or therapy. Most patients received corticosteroids (n = 7, 35%) and/or other immunosuppressive medications (n = 17, 85%) at the time of their MCR diagnosis. Other common risk factors for MCR in our cohort were diabetes mellitus (n = 7, 35%), malnutrition (n = 14, 70%), neutropenia (n = 7, 35%), and lymphopenia (n = 17, 85%).

Five out of the twenty patients with histology-proven pulmonary MCR grew mold, identified as *Mucor* (n = 3) or *Rhizopus* (n = 2) species. The majority (fifteen patients, 75%) with histology-proven pulmonary MCR were culture-negative. Univariate analysis of histopathological characteristics (Figure 2) did not reveal statistical differences between culture-positive and culture-negative cases. However, specimens from culture-positive patients exhibited a larger percentage of a moderate-to-extensive amount of necrosis on histopathology than culture-negative patients (100% vs. 67% of patients, *p* = 0.29), as well as a high burden of hyphae present (60% vs. 47% of patients with a moderate-to-extensive number of hyphae, *p* = 0.73). Conversely, culture-negative cases displayed a higher degree of vascular invasion than culture-positive patients (*p* = 0.09), with moderate-to-extensive vascular invasion seen in 64% and 50% of evaluable samples, respectively. In fact, extensive vascular invasion was only seen in specimens from culture-negative patients.

Although the differences investigated did not reach statistical significance, there were trends suggesting that culture-positive patients were more likely to have acute myeloid leukemia (60% vs. 27%, *p* = 0.19), a history of HCT (80% vs. 53%, *p* = 0.31), severe lymphopenia (absolute lymphocyte count ≤ 100/µL, 100% vs. 73%, *p* = 0.36), and monocytopenia (60% vs. 20%, *p* = 0.11) than culture-negative patients. Forty-two- and eighty-four-day all-cause mortality among all included patients with proven pulmonary MCR was 55% and 65%, respectively. The mortality of culture-positive and culture-negative patients was comparable through both day 42 (60% vs. 53%, *p* = 0.80) and day 84 (80% vs. 60%, *p* = 0.43).

## 4. Discussion

We reported the first pilot study comparing histopathological features and clinical outcomes to culture positivity in cases of histopathology-proven pulmonary MCR in patients with HMs and recipients of HCTs. Similar to prior studies that reported low rates of MCR culture positivity in this patient population, the percentage of culture-positive cases in our study was also small [2]. Since the early identification of MCR in clinical specimens is hindered by its poor recovery in culture, unveiling key histopathological characteristics associated with clinical outcomes and growth in culture can be important in managing these vulnerable patient populations. Moreover, identifying key histopathological characteristics in culture-negative cases also has value in identifying specimens that may require specialized laboratory conditions, as described in Kontoyiannis et al. [3], to promote the growth of stressed molds.

Predictably, we observed a higher burden of hyphae among culture-positive cases, which likely increased the probability of plating viable hyphae and achieving growth in culture. Additionally, more necrosis was also seen among culture-positive cases. Tissue necrosis may be a surrogate marker indicating more metabolically active molds that are more likely to grow under standard laboratory conditions. This observation might be linked to the toxin production of metabolically active Mucorales. For instance, the Mucoralean toxin mucoricin, which promotes host tissue necrosis, is predominantly produced by actively proliferating Mucorales and induced by encounters with host epithelia [6]. The percentage of cases displaying moderate-to-large tissue invasion was similar in both groups, but little-to-no tissue invasion was more commonly observed in the culture-negative cohort. Interestingly, we found a trend toward a higher degree of vascular invasion in specimens from culture-negative patients. We hypothesize that hyphae invading vessels are more likely to be exposed to higher concentrations of oxygen and host immune cells, possibly resulting in physiological stress and differences in carbon source and nutrient utilization, resulting in impedance of their growth under standard laboratory conditions. Ultimately, more cases are needed to confirm these intriguing observations.

This study has several limitations. Despite the long review period (>20 years), we only identified a limited number of patients that met the stringent inclusion criteria, resulting in poor statistical power to identify specific clinical or histopathological characteristics predicting culture positivity in cases of histology-proven pulmonary MCR. Additionally, our scoring methodology and the semiquantitative thresholds used for categorial classification have not been previously validated. Therefore, future in-depth studies ideally on multicenter, prospectively collected data, coupled with similar studies in experimental mammalian models of MCR, would be needed for independent validation of the methodology and further dissection of the nuanced histological parameters associated with culture positivity and/or clinical prognosis. Another limitation of these data is the reliance on the typical appearance of Mucorales in tissues as the diagnostic “gold standard” of MCR. However, there is significant interobserver variability in the accuracy of the histopathology-based identification of molds [7]. Specifically, in a multicenter study using culture as the diagnostic “gold standard”, 11% of culture-proven aspergillosis cases were misidentified as MCR by histopathology [8]. Conversely, 15% of tissue samples where the histopathologic diagnosis was consistent with hyalohyphomycetes grew Mucorales [9]. In fact, unusual Mucorales such as *Mucor velutinosus* have been described as *Aspergillus* in histopathology [10]. In addition, our study was grossly underpowered to evaluate whether there are differences among Mucorales in terms of culture positivity from infected lung tissues. Interestingly, a recent study indicated that some Mucorales, such as *Rhizimucor* species, do not grow well in clinical samples and are better diagnosed using PCR [11]. Lastly, nearly all patients in this study had cytopenia with significant cumulative immune suppression due to their underlying conditions. As such, the rates of histopathological characteristics observed in our patients cannot be extrapolated in other patient populations with MCR, such as patients with diabetes, patients with burns or trauma, or immunosuppressed but not myelosuppressed patients (e.g., non-neutropenic transplant recipients receiving corticosteroids).

## 5. Conclusions

This study represents the first comprehensive scoring method examining the relationship between histopathologic features, culture positivity, and clinical features among hematologic malignancy and HCT patients with histology-proven pulmonary MCR. Future endeavors are needed to validate this method based on multicenter data. Such studies will help define a combined score considering all four histopathological features in an aggregate point score that can be correlated with mortality, culture positivity, and fungal burden via quantitative PCR. Moreover, the application of our histopathological scoring methodology to animal models of MCR could provide a reference framework to standardize the notoriously difficult histopathological assessment of the pulmonary infection environment in preclinical studies of urgently needed novel therapeutic approaches to combat MCR.

## Figures and Tables

**Figure 1 jof-08-00380-f001:**
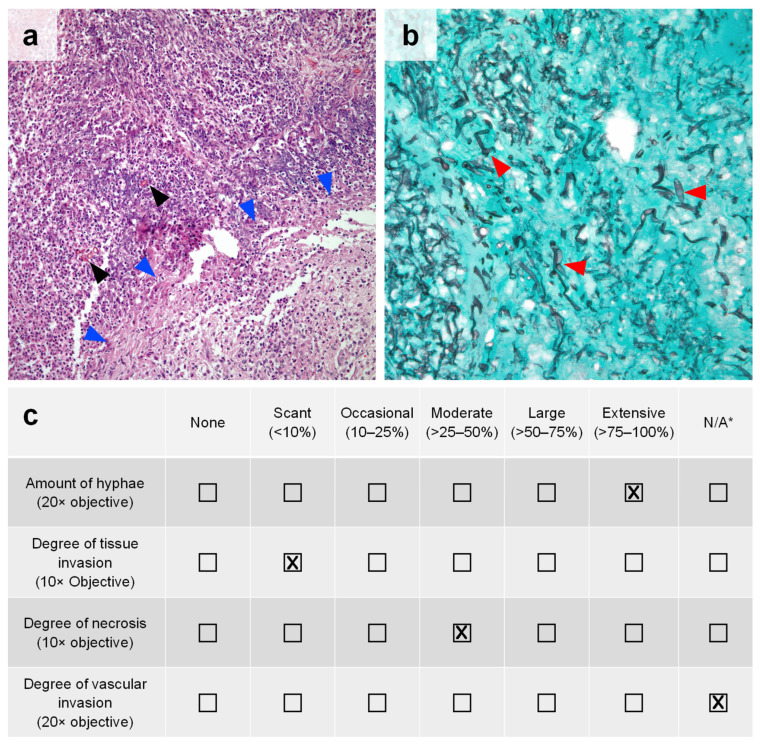
Pulmonary mucormycosis with histopathologic features of interest. (**a**) H&E staining of lung parenchyma with a moderate (>25–50%) amount of caseous necrosis (blue arrowheads) and acute suppurative inflammation plus scant (<10%) tissue invasion by MCR in cross-section (black arrowheads) at 10× objective. (**b**) GMS staining of a pulmonary tissue highlighting an extensive (>75–100%) amount of broad hyphae (red arrowheads) characteristic of MCR at 20× objective. No vessels are present on the slide to evaluate vascular invasion. (**c**) Example scorecard used for the semi-quantitative assessment of included pulmonary MCR cases based on the amount of histopathologic features present in the images shown in panels (**a**,**b**). * N/A = not applicable (histological feature was not present on the slide for the pathologist to evaluate).

**Figure 2 jof-08-00380-f002:**
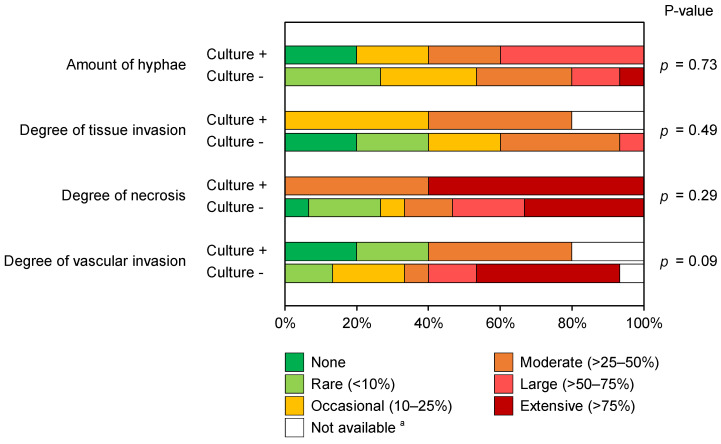
Stacked bar chart comparing the distributions of histopathological features of pulmonary mucormycosis in culture-positive and culture-negative patients. Mann–Whitney U test. ^a^ One case in the culture-positive cohort could not be evaluated on the degree of tissue invasion due to the lack of a corresponding GMS-stained slide. Additionally, two case specimens (one culture-positive and one culture-negative) did not contain vessels to evaluate vascular invasion.

**Table 1 jof-08-00380-t001:** Inclusion and exclusion criteria applied to histopathology specimens.

Inclusion Criteria	Exclusion Criteria
Specimen from pulmonary parenchyma	Case or slide lacking pulmonary parenchyma
Hyphae morphology present on slide consistent with mucormycosis	Mention or suggestion of alternative, non-Mucoralean mold hyphae morphology reported by pathology
Surgical (excisional or core needle biopsy) or autopsy specimens	Non-surgical, cytology, fine-needle aspirate, bronchial wash, or bronchoalveolar lavage specimens
Fungal culture sent from surgical or autopsy specimen/site	No corresponding fungal culture submitted from surgical or autopsy specimen/site

**Table 2 jof-08-00380-t002:** Univariate analysis of clinical characteristics in culture-positive versus culture-negative patients.

Characteristics, Median (Inter-Quartile Range) or N (%)	Total (N = 20)	Culture+ (N = 5)	Culture− (N = 15)	*p*-Value
Age, years ^a^	46.2 (25.5)	43.6 (28.5)	47.1 (27)	-
**Hematologic malignancy**				
Leukemia	16 (80)	4 (80)	12 (80)	>0.99
Acute myeloid leukemia	7 (35)	3 (60)	4 (26)	0.19
Myelodysplastic syndrome	1 (5)	0 (0)	1 (7)	0.94
Acute lymphoblastic leukemia	5 (25)	0 (0)	5 (33)	0.26
Chronic lymphocytic leukemia	1 (5)	0 (0)	1 (7)	0.94
Chronic myelogenous leukemia	2 (10)	1 (20)	1 (7)	0.41
Lymphoma	3 (15)	1 (20)	2 (13)	0.72
Multiple myeloma	1 (5)	0 (0)	1 (7)	0.94
**Hematopoietic cell transplant (HCT)**				
Any HCT	12 (60)	4 (80)	8 (53)	0.31
Allogeneic	11/12 (92)	4/4 (100)	7/8 (88)	0.74
Autologous	1/12 (8)	0/4 (0)	1/8 (13)	0.74
Acute or chronic GVHD in allogenic HCT recipients	7/11 (64)	2/4 (50)	5/8 (63)	0.79
**Cytopenia**				
Neutropenia ^a^	7 (35)	2 (40)	5 (33)	0.79
Absolute neutrophil count ≤100 cells/µL	6 (30)	2 (40)	4 (26)	0.58
Lymphopenia ^b^	17 (85)	5 (100)	12 (80)	0.48
Absolute lymphocyte count ≤100 cells/µL	16 (80)	5 (100)	11 (73)	0.36
Monocytopenia ^c^	6 (30)	3 (60)	3 (20)	0.11
**Other risk factors for mucormycosis ^d^**				
Recent antifungal prophylaxis or therapy	19 (95)	5 (100)	14 (93)	0.75
Glucocorticosteroids ^e^	7 (35)	2 (40)	5 (33)	0.79
Other immunosuppressive medications	17 (85)	4 (80)	13 (87)	0.72
History of diabetes mellitus	7 (35)	1 (20)	6 (40)	0.43
Hyperglycemia (>200 mg/dL)	4 (20)	1 (20)	3 (20)	>0.99
Malnutrition (serum albumin ≤ 3 g/dL)	14 (70)	4 (80)	10 (37)	0.58
Renal failure (serum creatinine > 2.5 mg/dL)	1 (5)	0 (0)	1 (7)	0.94
**Outcomes**				
All-cause mortality by 42 days	11 (55)	3 (60)	8 (53)	0.80
All-cause mortality by 84 days	13 (65)	4 (80)	9 (60)	0.43

Abbreviations and footnotes for Table 2. Abbreviation: GVHD = graft-versus-host disease. Footnotes: ^a^ Absolute neutrophil count ≤ 500 cells/µL; ^b^ absolute lymphocyte count ≤ 500 cells/uL; ^c^ absolute monocyte count ≤ 100 cells/µL; ^d^ at time of pulmonary mucormycosis diagnosis; ^e^ steroid use totaling ≥ 600 mg prednisone dose equivalent in past 30 days.

## Data Availability

Not applicable.

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
