# Peer review of "Taking a Closer Look: Clinical and Histopathological Characteristics of Culture-Positive versus Culture-Negative Pulmonary Mucormycosis"

_jof, 2022, doi:10.3390/jof8040380_

Round 1

Reviewer 1 Report

Useful pilot study that should be expanded in the future.

The only thing that I would like to see is a bit more information concerning culture. Specifically was the tissue ground or minced for culture? What were the choices of media, atmosphere and duration of incubation?

Author Response

Jof-1652492: Taking a closer look: Clinical and histopathological characteristics of culture-positive versus culture-negative pulmonary mucormycosis

Reviewer One)

“Useful pilot study that should be expanded in the future.

The only thing that I would like to see is a bit more information concerning culture. Specifically was the tissue ground or minced for culture? What were the choices of media, atmosphere and duration of incubation?”

Response to reviewer One)

We thank the reviewer for their suggestion to expand our methods section to include laboratory methods used in tissue processing and incubation for fungal culture of lung tissue. We have dedicated an additional subsection of the Materials and Methods section to address this.  Please see lines 79-91 (Materials and Methods subsection 2.3).

Reviewer 2 Report

The author present a 'pilot' study of the differences in histopathologic and clinical features of culture-positive and culture-negative mucormycosis. The strict inclusion and exclusion criteria resulted in only 20 patients in over a 20 year period.

The authors did not find a statistical difference in most of parameters compared although there was a trend towards a higher necrosis and hyphae burden in culture-positive cases and higher angio-invasive culture negative patients. The manuscript is well presented with appropriate tables and figures and the limitations and future recommendations are well discussed

The authors present a single study 'pilot' study comparing the clinical and histopathological features of culture positive mucormycosis and culture -negative mucormycosis. The study was conducted over 20 year and from the single center only 20 patients met the inclusion criteria. The study showed that although not statistically significant there was a trend for culture-positive patients to have a increased tendency of necrosis and high burden of hyphae but less vascular invasion compared to culture-negative patients with mycormycosis

The paper discusses the limitations of this study and make recommendations after for future studies. The paper was presented clearly and appropriate illustrations used.

Author Response

Reviewer Two)

“The author present a 'pilot' study of the differences in histopathologic and clinical features of culture-positive and culture-negative mucormycosis. The strict inclusion and exclusion criteria resulted in only 20 patients in over a 20 year period.

The authors did not find a statistical difference in most of parameters compared although there was a trend towards a higher necrosis and hyphae burden in culture-positive cases and higher angio-invasive culture negative patients. The manuscript is well presented with appropriate tables and figures and the limitations and future recommendations are well discussed”

“The authors present a single study 'pilot' study comparing the clinical and histopathological features of culture positive mucormycosis and culture -negative mucormycosis. The study was conducted over 20 year and from the single center only 20 patients met the inclusion criteria. The study showed that although not statistically significant there was a trend for culture-positive patients to have a increased tendency of necrosis and high burden of hyphae but less vascular invasion compared to culture-negative patients with mycormycosis

The paper discusses the limitations of this study and make recommendations after for future studies. The paper was presented clearly and appropriate illustrations used.”

Response to reviewer Two)

We thank the reviewer for their time and thoughtful comments with regard to evaluating our manuscript. We truly appreciate their contribution.

Reviewer 3 Report

Dear Authors,

The manuscript ID: jof-1652492_v1 entitled „Taking a closer look: Clinical and histopathological characteristics of culture-positive versus culture-negative pulmonary mucormycosis” written by Amy Spallone, Cesar A. Moran, Sebastian Wurster, Dierdre B. Axell-House and Dimitrios P. Kontoyiannis is very original.

Mucormycosis is a rare, emerging fungal infection, with high morbidity and mortality. Due to the rarity of the disease, it is almost impossible to conduct large, randomized clinical trials, and most of the available data regarding epidemiology, diagnosis, and treatment, originate from case reports and case series. This pilot study is very important as it compares the histopathological features and clinical outcomes with positive culture in cases of histopathologically confirmed lung mucormycosis in patients with hematologic malignancies and recipients of hematopoietic cell transplants.

The whole manuscript (Introduction, Materials and Methods, Results, Discussion, Conclusions) is properly organized. Introduction contains general data on mucormycosis. Appropriate methods were used to perform research. Results are documented. Statistical analysis was also performed. I think, it is a well written and original communication.

According to me, this manuscript is valuable and may be accepted for the publication in “Journal of Fungi”.

With highest regards,

Author Response

Reviewer Three)

“Dear Authors,

The manuscript ID: jof-1652492_v1 entitled „Taking a closer look: Clinical and histopathological characteristics of culture-positive versus culture-negative pulmonary mucormycosis” written by Amy Spallone, Cesar A. Moran, Sebastian Wurster, Dierdre B. Axell-House and Dimitrios P. Kontoyiannis is very original.

Mucormycosis is a rare, emerging fungal infection, with high morbidity and mortality. Due to the rarity of the disease, it is almost impossible to conduct large, randomized clinical trials, and most of the available data regarding epidemiology, diagnosis, and treatment, originate from case reports and case series. This pilot study is very important as it compares the histopathological features and clinical outcomes with positive culture in cases of histopathologically confirmed lung mucormycosis in patients with hematologic malignancies and recipients of hematopoietic cell transplants.

The whole manuscript (Introduction, Materials and Methods, Results, Discussion, Conclusions) is properly organized. Introduction contains general data on mucormycosis. Appropriate methods were used to perform research. Results are documented. Statistical analysis was also performed. I think, it is a well written and original communication.

According to me, this manuscript is valuable and may be accepted for the publication in “Journal of Fungi”.

With highest regards,”

Response to reviewer Three)

We thank the reviewer for their thorough review of our manuscript and their thoughtful commentary. We truly appreciate their time and feedback.

Reviewer 4 Report

Good paper, novel approach.  Will help us understand better mould biology in immunocompromised persons.

Line 66

This paragraph contains many of the definitions used by the authors.  However “time of diagnosis” is not defined, though it appears to be a crucial parameter, since other definitions stem from it (eg, antifungal use in the 30 days before time of diagnosis).  I have encountered different definitions in the literature – day biopsy was performed, first day of fever, day if the suggestive/characteristic imaging, day the bug grew on culture …

Line 139

Paragraph beginning line 139 needs to be re-phrased completely. Since none of the differences reached statistical significance, the phrase “more likely to have” is inappropriate. A carefully phrased sentence might read like this: “Although the differences did not reach statistical significance, there were trends suggesting that …”  Other phrases might include “tended towards” or “there appeared to be more”.

Line 180               This sentence is hanging – “Therefore, future in-depth studies on multicentre”

Author Response

Reviewer Four)

“Comments and Suggestions for Authors

Good paper, novel approach.  Will help us understand better mould biology in immunocompromised persons.

Line 66

This paragraph contains many of the definitions used by the authors.  However “time of diagnosis” is not defined, though it appears to be a crucial parameter, since other definitions stem from it (eg, antifungal use in the 30 days before time of diagnosis).  I have encountered different definitions in the literature – day biopsy was performed, first day of fever, day if the suggestive/characteristic imaging, day the bug grew on culture …

Line 139

Paragraph beginning line 139 needs to be re-phrased completely. Since none of the differences reached statistical significance, the phrase “more likely to have” is inappropriate. A carefully phrased sentence might read like this: “Although the differences did not reach statistical significance, there were trends suggesting that …”  Other phrases might include “tended towards” or “there appeared to be more”.

Line 180               This sentence is hanging – “Therefore, future in-depth studies on multicentre”

Response to reviewer Four)

Line 66) We thank the reviewer for their request for clarification on the definition of date of diagnosis. For the purposes of our study, date of diagnosis was recorded as the date of collection of histopathology that revealed hyphae laden tissue consistent with mucormycosis. We have incorporated this clarification in lines 67-68.

Line 139) We thank the reviewer for their assistance with improving the accuracy of our manuscript. We have incorporated their feedback on lines 139, which can be found at line 155-159.

Line 180) We thank the reviewer for catching this typo. The following correction to line 180 has been made and can be found at lines 196-200.

“Therefore, future in-depth studies on multicenter, ideally, prospectively collected data, coupled with similar studies in experimental mammalian models of MCR, would be needed for independent validation of the methodology and further dissection of the nuanced histological parameters associated with culture positivity and/or clinical prognosis.”